# Ventricular Arrhythmias in Patients with Implanted Cardiac Devices at High Risk of Obstructive Sleep Apnea

**DOI:** 10.3390/medicina58060757

**Published:** 2022-06-02

**Authors:** Akram Khan, Ryan D. Clay, Asha Singh, Chitra Lal, Larisa G. Tereshchenko

**Affiliations:** 1Division of Pulmonary, Allergy & Critical Care Medicine, Oregon Health & Science University, Portland, OR 97239, USA; ryan.d.clay@kp.org; 2Department of Neurology, Oregon Health & Science University, Portland, OR 97239, USA; singas@ohsu.edu; 3Division of Pulmonary, Critical Care, Allergy and Sleep Medicine, Medical University of South Carolina, Charleston, SC 29425, USA; lalch@musc.edu; 4Department of Quantitative Health Sciences, Lerner Research Institute, Cleveland Clinic, 9500 Euclid Ave JJN3-01, Cleveland, OH 44195, USA; tereshl@ccf.org

**Keywords:** sleep apnea syndromes, arrhythmias, cardiac, epidemiology, surveys and questionnaires

## Abstract

*Background and Objectives*: Patients with pre-existing cardiac disease have a higher prevalence of Obstructive Sleep Apnea (OSA). OSA has been associated with an increased risk of supraventricular and ventricular arrhythmia. We screened subjects with implanted pacemakers and automated implantable cardioverter defibrillators (AICD) for OSA with the Berlin Questionnaire and compared the incidence of ventricular arrhythmias and automated implantable cardioverter defibrillator (AICD) firing between high and low OSA risk groups. *Materials and Methods*: We contacted 648 consecutive patients from our arrhythmia clinic to participate in the study and performed final analyses on 171 subjects who consented and had follow-up data. Data were abstracted from the electronic health record for the incidence of non-sustained ventricular tachycardia (NSVT), ventricular tachycardia (VT), ventricular fibrillation (VF) and AICD firing and then compared between those at high versus low risk of OSA using the Berlin Questionnaire and multivariate negative binomial regression. *Results*: The average follow-up period was 24.2 ± 4.4 months. After adjusting for age, gender and history of heart failure, those subjects at high risk of OSA had a higher burden of NSVT vs. those with a low risk of OSA (33.4 ± 96.2 vs. 5.82 ± 17.1 episodes, *p* = 0.003). A predetermined subgroup analysis of AICD recipients also demonstrated a significantly higher burden of NSVT in the high vs. low OSA risk groups (66.2 ± 128.6 vs. 18.9 ± 36.7 episodes, *p* = 0.033). There were significant differences in the rates of VT, VF or AICD shock burden between the high and low OSA risk groups and in the AICD subgroup analysis. *Conclusions*: There was increased ventricular ectopy among pacemaker and AICD recipients at high risk of OSA, but the prevalence of VT, VF or AICD shocks was similar to those with low risk of OSA.

## 1. Introduction

Obstructive sleep apnea (OSA) affects up to 24% of men and 9% of women [1,2,3] and has been associated with an increased risk of cardiovascular disease as well as mortality [3,4,5]. Patients with OSA have a higher incidence of a variety of arrhythmias [6,7,8,9,10,11,12,13]. We wanted to better understand the association between the risk of OSA and ventricular arrhythmias so we classified patients with pre-existing cardiovascular disease in our arrhythmia clinic into high or low risk of OSA groups using the Berlin Questionnaire to compare the prevalence of ventricular arrhythmias and automatic implantable cardioverter defibrillator (AICD) firings between the two risk groups. We hypothesized that there would be increased ventricular arrhythmias and AICD firing among the high-risk group.

## 2. Materials and Methods

After IRB approval, we reviewed data on 648 consecutive patients seen between 2016 and 2018 in our arrhythmia clinic, a population at higher risk of ventricular arrhythmias that routinely reported arrhythmias according to device data downloaded during follow-up visits. Patients were eligible if they had implanted pacemakers or AICDs, had a follow-up at our institution and were between 18–99 years old. Investigators contacted patients by phone for informed consent and either verbally administered or mailed the Berlin Questionnaire for completion. The Berlin Questionnaire is a validated 10-item survey that takes body mass index (BMI), comorbid hypertension and symptoms of OSA into account to stratify patients into having a high or low risk of OSA [14]. Of the 648 subjects who were contacted, 242 agreed to participate. Sixty-six subjects had to be excluded due to a lack of follow-up data on the AICD or pacemaker. Two subjects were excluded because of inconsistent data, and three subjects were excluded because they had implantable loop recorders, not AICDs or pacemakers. A final analysis was completed on 171 subjects (Figure 1).

Data were retrospectively abstracted to allow for a 24-month follow-up, including a follow-up using a remote device. Arrhythmias were detected by internal device algorithms, adjudicated during the clinic visit by treating clinicians, and educated and reviewed by authors if uncertain (RC, AK). Arrhythmias, including non-sustained ventricular tachycardia (NSVT), ventricular tachycardia (VT), ventricular fibrillation (VF) and AICD firing, were counted per patient over the follow-up period. All AICD firings were counted regardless of whether they were determined appropriate or inappropriate therapy. One investigator (RC) audited data abstraction and reconciled discrepancies with the help of the principal investigator (AK). Comorbid diagnoses including sleep apnea were abstracted from the electronic health record (EHR). Data on the apnea-hypopnea index and continuous positive airway pressure (CPAP) or bi-level positive airway pressure (BPAP) compliance were not available for abstraction.

## 3. Statistical Analysis

Statistical analysis was carried out using Stata 14 (Stata Corp, College Station, TX, USA). Continuous data were analyzed using a two-sided *t*-test and categorical data using a Chi-square test. Fisher’s exact test was used for categorical data when there were five or fewer counts in a category. When non-numeric values such as “occasional” or “rare” were recorded in EHR to quantify episodes of non-sustained ventricular tachycardia (NSVT), we recorded a numerical value of “1” given that we could not quantify the number of episodes.

The data followed a dispersed distribution with a standard deviation greater than the mean in both the high and low OSA risk groups, when arrhythmia incidence was analyzed. Since our data were in the form of count variables, we modeled our data using negative binomial regression to look for a significant difference in arrhythmia counts over the follow-up period using a coefficient of correlation, which measures the difference in the logs of expected counts of the response variable for a one-unit change in the predictor variables, given that the other predictor variables in the model are held constant [15,16,17]. Covariates were analyzed by univariate binomial regression to look for associations to build our final model, which included age and gender. Due to the lack of time-to-event data, we analyzed VT, VF, NSVT and AICD firings as cumulative counts to compare means between the high and low OSA risk groups. This was then confirmed with zero-inflated negative binomial regression, adjusting for values with zero counts. Data analysis was also completed in a predetermined group of AICD recipients adjusting for age, gender, and history of CHF. Results were also adjusted for follow-up exposure.

## 4. Results

In our cohort of 171 subjects with pacemakers and AICDs, the average age was 64.6 ± 16 (median, IQR 67, 55–77) years and BMI 28.5 ± 7.1 (median, IQR 27.4, 23.6–31.9) kg/m^2^. The follow-up period averaged 24.2 months (95% CI 23.6–24.9) with a range from 6 to 39 months with no significant influence after adjusting for follow-up exposure. Seventy-three (42.7%) of our subjects experienced ventricular rhythm abnormalities (NSVT, VT or VF) over the follow-up period. The majority of ventricular arrhythmias were NSVT with a mean of 20.2 ± 71.6 episodes of NSVT, 1.8 ± 8.1 episodes of VT and 0.1 ± 0.6 episodes of VF per subject over the follow-up period.

Eighty-two subjects (48%) had a low risk and 89 subjects (52%) had a high risk of OSA based on the Berlin Questionnaire. There was a significantly higher prevalence of atrial fibrillation in the low-risk group and a significantly higher number of AICD recipients in the high-risk group (Table 1). There were no significant differences among the baseline characteristics and comorbidities between the two groups (Table 1). BMI is part of the Berlin Questionnaire and was expected to be significantly higher in the high OSA risk group: 30.1 (95% CI 25.4–28.1) versus 26.7 (95% CI 28.5–31.8 *p* = 0.02) kg/m^2^. In view of low event counts, ventricular tachycardia (VT) and ventricular fibrillation (VF) episodes were summed together for each subject over the follow-up period to generate the category “high-risk ventricular rhythms” and analyzed as a separate category.

The prevalence of ventricular arrhythmias in our population was low (Table 2, Figure 2a–d). Using negative binomial regression, there was no significant difference between mean VT or VF episodes in the high versus low OSA risk groups or the combined category of high-risk ventricular rhythms on univariate analysis. There was a significant increase in the NSVT rate in the high-risk Berlin category (coef. 1.75, *p* = 0.0009, 95% CI 0.80–2.69) (Table 2) by univariate negative binomial regression.

In the univariate analysis, only a history of heart failure and having an AICD in place were significantly associated with a risk of OSA, based on the Berlin Questionnaire. We developed a multifactorial model to analyze the association between the risk of OSA and ventricular arrhythmias using negative binomial regression. This included adjustments for the diagnosis of heart failure, which was significantly associated with NSVT in the univariate analysis, and adjustments for age and gender, as they can both influence the risk of OSA [18]. After adjusting for age, gender and heart failure, a high risk of OSA was independently associated with a 314% higher rate of NSVT (coef. 1.42, *p* = 0.003, 95% CI 0.50–2.35). The risk of OSA also increased by 18.8% per 5-year increase in age (*p* 0.03) and 429% with a history of heart failure (coef. 1.84, *p* = 0.001, 95% CI 0.76–2.93). 

There were more AICD recipients in the high-risk group, and AICD-recipient status was an independently significant covariate (coef. 4.08, *p* < 0.001, 95% CI 3.2–4.89) for the risk of NSVT. To explore this, we performed a subgroup analysis of the 72 AICD recipients (Table 3). The average age for the AICD subjects was 61.6 ± 15.1 years, and they were followed for an average of 24.9 ± 3.3 months. There were no significant differences in mean NSVT, VT, VF or ICD discharge between the high and low OSA risk groups in AICD recipients (Table 4). However, in this predetermined subgroup analysis, we again noted significantly higher odds of having NSVT among the high OSA risk group, when compared with the low OSA risk group, with a mean of 62.3 episodes per subject versus 17.6 episodes over the follow-up period and a 256% higher rate of NSVT over the follow-up period (coef. 1.27, *p* = 0.025, 95% CI 0.16–2.37). This held true after adjusting for age, gender, and heart failure among the 72 ACID recipients in negative binomial regression (207% increase in NSVT rate, coef. 1.21, *p* = 0.033, 95% CI 0.10–2.33).

Data on ejection fraction were available for 66 of the 72 subjects with AICD. We stratified these subjects into those with an EF ≤ 35% and those with an EF > 35%, based on guidelines for the implantation of AICD for the primary prevention of ventricular arrhythmia [19,20]. The number of subjects with an EF of less than 35% was not statistically different between the high and low risk OSA groups. An ejection fraction of less than 35% was not a significant predictor of ventricular arrhythmia among the AICD recipients (coef. 0.43, *p* = 0.49, 95% CI −0.81 to 1.68) in univariate analysis or with Berlin risk category. As expected, having an EF < 35% predicted AICD firing. There were only five episodes of inappropriate AICD firing in the study.

We extracted a history of sleep apnea and prescription for home positive airway pressure (PAP) therapy from EHR; however, AHI and NIPPV compliance data were not available. Excluding patients diagnosed with sleep apnea or prescribed home NIPPV did not change the association between high-risk Berlin status and NSVT.

## 5. Discussion

Association studies have shown a link between OSA and ventricular and supraventricular arrhythmias [3,8,9,11,12,21,22]. Patients with untreated OSA undergoing ablation for atrial fibrillation have higher rates of recurrence compared with those on appropriate CPAP therapy [21]. The DREAM study investigators found a 4-fold increase in the risk of supraventricular tachycardia in patients with OSA [22]. Similarly, a study of patients with hypertrophic obstructive cardiomyopathy showed that the presence of OSA was associated with NSVT [9]. In patients with heart failure, sleep apnea was associated with atrial fibrillation in the DASAP-HF study [11]. The association of OSA with ventricular arrhythmias is less clear, although treatment of sleep apnea has been associated with a decreased risk of ventricular arrhythmias [10]. A small number of studies evaluated OSA in patients with implanted cardiac devices. The European Multicenter Polysomnographic Study showed that 59% of patients with long-term pacing had OSA on polysomnography [13]. Other studies demonstrated a similarly high incidence of OSA among pacemaker and AICD recipients, ranging from 40–66% [11,23,24,25,26,27,28,29]. In a study of patients with ejection fractions less than 35%, and with AICD similar to data from our study, there were no significant ventricular arrhythmias or worsening of heart failure; however, there was a slightly increased incidence of atrial fibrillation [11].

We screened subjects with pacemakers or AICD for the risk of OSA using the Berlin Questionnaire and compared the incidence of ventricular arrhythmia and AICD firing between the high and low OSA risk groups over a 2-year period. We found that, similar to prior studies, 52.1% of the study population and 61.1% of the ACID subgroup had a high risk of OSA using the Berlin Questionnaire [1,2,11]. Adjusting for age, gender and a diagnosis of heart failure, we found significantly more NSVT among patients in the high OSA risk category. This association persisted when the AICD subgroup was analyzed separately, similar to the study by Boatani et al. [11] We did not find significant differences in the incidence of VT, VF or AICD discharge between the low and high OSA risk groups. On combining VF and VT as high-risk ventricular rhythms, 22.5% of the patients in the high OSA risk group experienced high-risk ventricular rhythms compared with 13.4% of the low-risk group during the follow-up period, although this difference did not meet statistical significance. The increased ventricular ectopy among the high OSA risk individuals may suggest underlying cardiac irritability from OSA and a need to screen patients with pacemakers and AICD for sleep apnea and provide appropriate therapy. Our study included subjects with a prior diagnosis of sleep apnea and/or use of NIPPV; however, due to the retrospective nature of our analysis, compliance with NIPPV was not available. Excluding these patients from the analysis did not change the results.

Current evidence is unclear as to whether OSA is associated with an increased risk of ventricular arrhythmia among patients with cardiac disease [3,11]. Neither Fries nor Padeletti found significant differences in ventricular arrhythmia between device recipients with diagnosed sleep apnea, compared to those without sleep apnea [26,27]. However, Zeidan-Shwiri and Serizawa both found that patients with sleep apnea received significantly more appropriate AICD therapies and that these events more often occurred during sleep, presumably when apneas and hypopneas were occurring [28,29]. Additionally, the DREAM study showed increased odds of cardiac arrhythmia at night in patients with severe OSA [22]. Bitter found that patients with OSA had a shorter duration of time until their first appropriate AICD therapy compared to patients without OSA [24]. Smaller studies suggest that PAP therapy reduces ventricular irritability in heart failure patients with OSA and in OSA patients with pre-existing arrhythmia [9,30,31]. Given the link between arrhythmia and OSA, it was surprising to find a higher prevalence of atrial fibrillation in the low OSA risk group. Our study center is a tertiary referral center, and this may have been biased toward specialized procedures for atrial fibrillation management.

Our population had a high prevalence of heart failure, and thus a lower cardiovascular reserve to tolerate the maladaptive physiology of OSA, likely providing a substrate for increased ventricular ectopy and increased NSVT in the high OSA risk group. While the literature shows that NSVT is associated with increased mortality in patients with underlying cardiovascular disease, current antiarrhythmic therapies have not been able to decrease mortality [32,33]. We speculate that appropriate PAP therapy may be able to fill this therapeutic gap based on data from smaller studies [10,12,31].

Our study has several limitations related to being a retrospective cohort study. Our sample size is small, and the lack of PSG data makes the classification of sleep apnea status less accurate. Additionally, only 37% of contacted individuals agreed to participate in the study, which could potentially introduce bias. It is also possible that the low incidence of high-risk ventricular rhythms and AICD firings in our population may have made it difficult to find a difference between the high and low OSA risk groups. Data on the treatment compliance for those undergoing PAP for OSA was unavailable, which may also bias our results toward the null hypothesis. Unfortunately, we did not have data available that allowed us to analyze time to arrhythmia between the high and low OSA risk groups, which may be a more sensitive markers than cumulative arrhythmia events. Additionally, we did not analyze subjects’ home medications, which can affect both sleep architecture as well as arrhythmias.

Despite the limitations of our study, we demonstrated a substantially higher risk of OSA in patients with known cardiac disease and found that ventricular ectopy was increased among those with cardiac disease, with a high likelihood of comorbid OSA. We were unable to explore this further due to difficulties in obtaining polysomnography data in our retrospective cohort but feel that this shows the need for OSA screening and follow-up among this population. Our data suggest a high prevalence of OSA in patients with cardiovascular disease, with an even higher prevalence in the AICD subgroup. There is a higher burden of ventricular ectopy in the group at high risk of OSA. Further studies are needed to define the prevalence of OSA in this population using in-lab or home sleep testing and to determine whether treatment with PAP would decrease ventricular ectopy and benefit mortality.

## Figures and Tables

**Figure 1 medicina-58-00757-f001:**
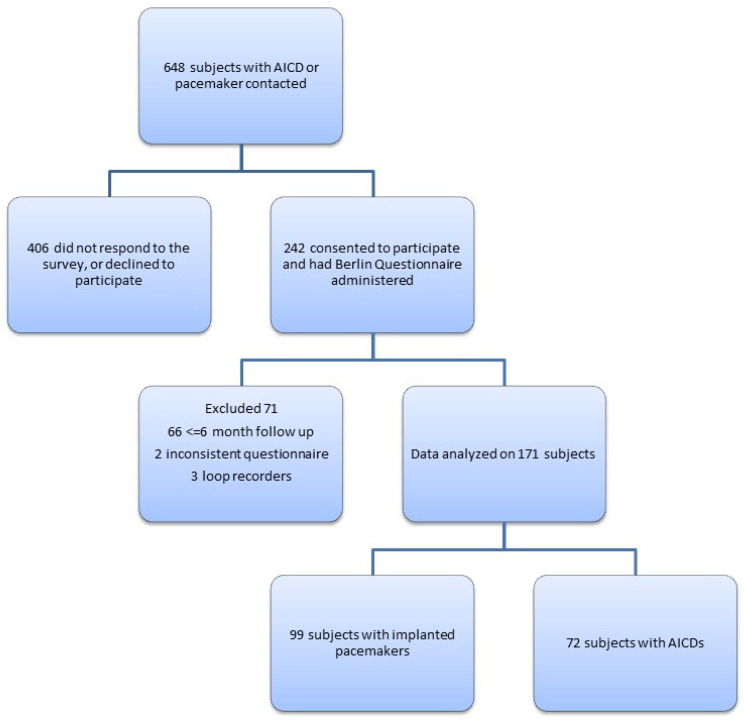
Study population selection.

**Figure 2 medicina-58-00757-f002:**
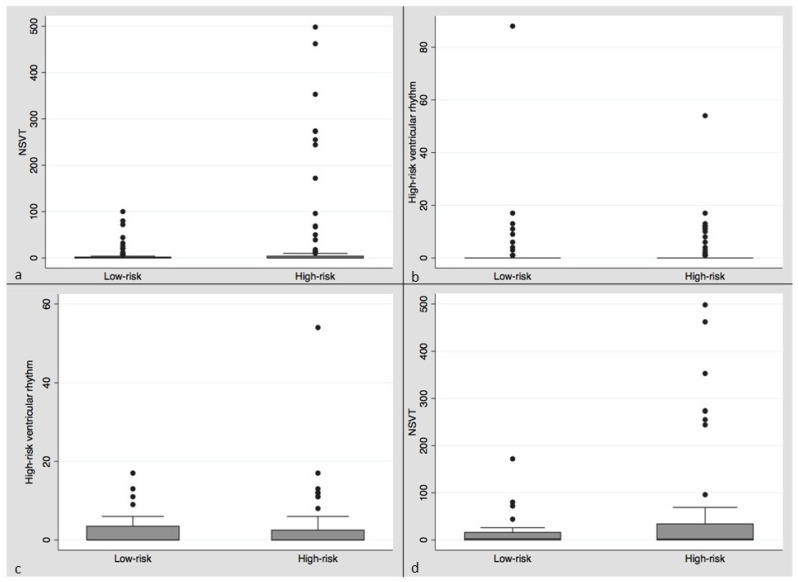
Box plots (mean, median, IQR) clockwise from upper left: (**a**) Box plot of NSVT episodes per patient in low versus high OSA risk categories over the follow-up period. (**b**) Box plot of high-risk ventricular rhythm episodes per patient in low versus high OSA risk categories over the follow-up period. (**c**) Box plot of NSVT episodes per patient in low versus high OSA risk categories in the AICD subgroup over the follow-up period. (**d**) Box plot of high-risk ventricular rhythms per patient in the low versus high OSA risk categories in the AICD subgroup over the follow-up period.

**Table 1 medicina-58-00757-t001:** Baseline characteristics of 171 subjects.

	Low Risk of OSA on Berlin Questionnaire	High Risk of OSA on Berlin Questionnaire	*p* Value
	n = 82 (48.0%)	n = 89 (52.0%)	
Age (years)	65.9 (±16.9)	63.5 (±15.2)	0.315
BMI (kg/m^2^)	26.7 (±6.1)	30.1 (±7.6)	0.002
Mean follow-up (months)	24.4 (±4.1)	24.1 (±4.7)	0.672
Gender			
Male	43 (52.4%)	49 (55.1%)	0.732
Female	39 (47.6%)	40 (44.9%)	0.732
Race			
White	80 (97.6%)	84 (94.4%)	0.246
Black	0 (0%)	3 (3.4%)	0.246
Asian	2 (2.6%)	1 (1.1%)	0.246
Native American	0 (0%)	1 (1.1%)	0.246
Ethnicity			
Non-Hispanic	80 (97.6%)	86 (96.6%)	0.718
Hispanic	2 (2.6%)	3 (3.4%)	0.718
On home PAP therapy at baseline	22 (26.8%)	22 (32.8%)	0.752
On home oxygen at baseline	6 (7.3%)	7 (7.9%)	0.893
CAD	27 (32.9%)	41 (46.1%)	0.079
HTN	41 (50%)	51 (57.3%)	0.339
DMII	18 (22.0%)	26 (29.2%)	0.278
CKD	12 (14.6%)	16 (18.0%)	0.555
COPD	6 (7.3%)	14 (15.7%)	0.087
A fib/flutter	44 (53.7%)	34 (38.2%)	0.043
Stroke	8 (9.8%)	12 (13.5%)	0.433
Sleep apnea	23 (28.1%)	28 (31.5%)	0.626
Heart failure	35 (42.7%)	49 (55.1%)	0.106
ICD in place	33 (40.1%)	52 (58.4%)	0.018
Valvular Heart Disease	12 (14.6%)	21 (23.6%)	0.138

BMI: body mass index, CAD: coronary artery disease, HTN: hypertension, DMII: diabetes mellitus Type II, CKD: chronic kidney disease, COPD: chronic obstructive lung disease, LVAD: left ventricular assist device, ICD: implantable cardioverter-defibrillator, PAP: positive airway pressure. *p* value: probability value.

**Table 2 medicina-58-00757-t002:** Frequency of ventricular arrhythmias.

	Low-Risk Berlin (n = 82)	High-Risk Berlin (n = 89)	*p* Value
VT episodes (mean)	1.8 (±9.6)	1.9 (±6.4)	0.95
VF episodes (mean)	0.1 (±0.5)	0.2 (±0.7)	0.32
NSVT episodes (mean)	5.8 (±17.1)	33.4 (±96.2)	0.01
High-risk ventricular rhythms (mean)	1.9 (±10.1)	2.0 (±6.7)	0.90

VT: ventricular tachycardia, VF: ventricular fibrillation, NSVT: non-sustained ventricular tachycardia. High-risk ventricular rhythms = VT and VF summed. p value: probability. Determined by 2-sided *t*-test.

**Table 3 medicina-58-00757-t003:** Baseline characteristics of AICD subgroup.

	Low-Risk Berlin	High-Risk Berlin	*p* Value
	n = 28 (37.8%)	n = 44 (61.1%)	
Age (years)	64.5 (±13.8)	59.7 (±15.7)	0.186
BMI (kg/m^2^)	27.8 (±6.9)	30.9 (±7.7)	0.087
Follow-up (months)	25.1 (±2.0)	24.8 (±3.9)	0.695
Gender			
Male	19 (67.9%)	26 (59.1%)	0.454
Female	9 (32.1%)	18 (40.9%)	0.454
On home PAP therapy at baseline	7 (25.0%)	10 (22.7%)	0.825
On home oxygen at baseline	4 (14.3%)	5 (11.4%)	0.715
CAD	11 (39.3%)	21 (47.7%)	0.482
HTN	13 (46.4%)	20 (45.5%)	0.936
DMII	8 (28.6%)	14 (31.2%)	0.771
CKD	5 (17.9%)	11 (25%)	0.477
COPD	2 (7.14%)	6 (13.6%)	0.393
A fib/flutter	14 (50.0%)	13 (29.6%)	0.081
Stroke	1 (3.9%)	6 (13.6%)	0.160
Sleep apnea	9 (32.1%)	14 (31.8%)	0.977
Heart failure	21 (75.0%)	33 (75.0%)	1
EF < 35% (n = 66) *	12 (46.2%)	18 (45%)	0.93
Valvular Heart Disease	3 (10.7%)	10 (22.7%)	0.196

BMI: body mass index, CAD: coronary artery disease, HTN: hypertension, DMII: diabetes mellitus Type II, CKD: chronic kidney disease, COPD: chronic obstructive lung disease, LVAD: left ventricular assist device, ICD: implantable cardioverter defibrillator, PAP: positive airway pressure. * calculated by Chi-square. Data on EF only available for 66 subjects.

**Table 4 medicina-58-00757-t004:** Ventricular arrhythmia in AICD subgroup determined by 2-sided *t*-test.

	Low-Risk Berlin (n = 28)	High-Risk Berlin (n = 44)	*p* Value
VT episodes (mean)	2.4 (±4.4)	3.5 (±8.9)	0.54
VF episodes (mean)	0.1 (±0.3)	0.3 (±0.8)	0.31
NSVT episodes (mean)	17.6 (±36.7)	62.3 (±128.6)	0.08
High-risk ventricular rhythms (mean)	2.5 (±4.6)	3.8 (±9.1)	0.49
ICD shocks (mean)	0.3 (±0.7)	0.5 (±1.8)	0.48

VT: ventricular tachycardia, VF: ventricular fibrillation, NSVT: non-sustained ventricular tachycardia. High-risk ventricular rhythm = VT and VF summed.

## Data Availability

Data available on request due to restrictions patient privacy. The data presented in this study are available on request from the corresponding author. The data are not publicly available due to HIPPA.

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
