# Peer review of "Ventricular Arrhythmias in Patients with Implanted Cardiac Devices at High Risk of Obstructive Sleep Apnea"

_medicina, 2022, doi:10.3390/medicina58060757_

Round 1

Reviewer 1 Report

the authors assessed in the present article the rate of ventricular arrhytmias in high and low OSA-risk patients with implanted cardiac devices.

overall, the article has been well presented, language is quite strait-forward. references are adequate to back their results up.

my conderns regarding the article are;

1- it seems that around 30 % of the two subgroups (namely, the low- and high-OSA risk) possesses OSA for real. However, data regarding their respective OSA significance has not been presented and , in my opinion, this 30% of the patients in high-OSA risk subgroup are very likely to have severe OSA described by polysomnography, as compared to those in low-OSA risk subgroup. What do the authors consider about this? Higher insidance of NSVT in high OSA-risk subgroup may be biased due to this possibility?  

2- it would be perfect if the authors add to the article the time interval that most of the NSVT burden in high OSA-risk subgroup has been encountered, namely, at night? or in the morning? or homogenous distribution throuout the daytime?

3- please replase EF with LVEF 

Author Response

1- it seems that around 30 % of the two subgroups (namely, the low- and high-OSA risk) possesses OSA for real. However, data regarding their respective OSA significance has not been presented and in my opinion, this 30% of the patients in high-OSA risk subgroup are very likely to have severe OSA described by polysomnography, as compared to those in low-OSA risk subgroup. What do the authors consider about this? Higher incidence of NSVT in high OSA-risk subgroup may be biased due to this possibility?  

Response:

We thank the reviewer for pointing this out. Unfortunately, while all patients had pacemakers and ICD’s were placed at our institution, sleep studies were not done at our institution and AHI and sleep study data was only available for 4 subjects in the low risk group (AHI = 29 +/- 21.5) and 9 subjects in high risk group (AHI = 34.1 +/- 27.2). The p value was 0.74 and the difference was not statistically significant. We reanalyzed data after removing patients with know sleep apnea and the results were similar with no changes. The sample size was too small to report the data for meaningful conclusions.

2- it would be perfect if the authors add to the article the time interval that most of the NSVT burden in high OSA-risk subgroup has been encountered, namely, at night? or in the morning? or homogenous distribution throughout the daytime?

Response: This is also a very important point. We tried to look for the data needed. Time interval for arrythmias was not available in ICD downloads and was not collected for this analysis. We agree with the reviewer that such data would be very useful and plan to collect it in future prospective research projects for which we plan to apply for funding and use this project as preliminary data.

3- Please replace EF with LVEF.

Response: thank you for suggesting this clarification . This has been completed and EF has been replaced with LVEF (Left ventricular ejection fraction).

Reviewer 2 Report

- I believe that the authors should provide information about proper sample size calculation, and if this has not been done, it should be added to the limitations of the study in the discussion section of the manuscript.

- I think that the discussion section of the manuscript should start with the paragraph, which is now second in its current form.

- I think that the findings of previous studies, which are discussed in the 1st and 3rd paragraph of the discussion section of the manuscript in its current form, should be presented in one paragraph and with a clearer comparison with the findings of the present study.

- I think that the authors should elaborate further on the implications of the findings of the present study in the management of patients with ventricular arrythmias and high risk of OSA, and on the future research that has to be done on this field.

Author Response

- I believe that the authors should provide information about proper sample size calculation, and if this has not been done, it should be added to the limitations of the study in the discussion section of the manuscript.

Response: Thank you for suggesting this limitation. This is a retrospective case-control study so sample size calculations were not possible. We thank the reviewers of this comment, and a statement has been added to the limitations section of the manuscript.

- I think that the discussion section of the manuscript should start with the paragraph, which is now second in its current form.

Response: We want to thank the reviewer for this comment, we have moved the paragraph as suggested and the manuscript now reads much better than before! This insight and input is very deeply appreciated.

- I think that the findings of previous studies, which are discussed in the 1st and 3rd paragraph of the discussion section of the manuscript in its current form, should be presented in one paragraph and with a clearer comparison with the findings of the present study.

Response: thank you for the suggestion. We have moved that initial 1st and 3rd paragraph 2 below where the reviewer suggested. We have combined paragraph 1 & 3 to a single paragraph and added comparisons with our data when possible.

- I think that the authors should elaborate further on the implications of the findings of the present study in the management of patients with ventricular arrythmias and high risk of OSA, and on the future research that has to be done on this field.

Response: In the terminal segment of the paper, we have further elaborated the implications of the findings of our study and clarified areas for future research.

Round 2

Reviewer 2 Report

Good work!

Congratulations!